# Development of Novel Blackgram (*Vigna mungo* (L.) Hepper) Mutants and Deciphering Genotype × Environment Interaction for Yield-Related Traits of Mutants

**Manickam Dhasarathan** [1,*], **Seshadri Geetha** [1], **Adhimoolam Karthikeyan** [2], **Datchinamoorthy Sassikumar** [3] **and Narayanapillai Meenakshiganesan** [4]

1. Centre for Plant Breeding and Genetics, Tamil Nadu Agricultural University, Coimbatore 641003, India; geethagovind1@gmail.com
2. Subtropical Horticulture Research Institute, Jeju National University, Jeju City 690-756, Korea; karthik2373@gmail.com
3. Tamil Nadu Rice Research Institute, Aduthurai 612101, India; sassiku@rediffmail.com
4. Department of Genetics and Plant Breeding, Tamil Nadu Agricultural University, Coimbatore 641003, India; meenakshignsn@yahoo.co.in
* Correspondence: dhasa5589@gmail.com

**Abstract:** Blackgram (*Vigna mungo* (L.) Hepper) yields are noticeably poor due to a shortage of improved varieties and an aggravated narrow genetic base. An attempt was made to isolate novel blackgram mutants by selecting for yield-related traits derived through gamma irradiation and testing the mutant genotype's stability across the different environments. The irradiated blackgram populations $M_1$-$M_5$ were established in the background of cultivars ADT 3, Co 6, and TU 17-9. Desirable mutants were selected from $M_3$ to $M_5$ generations. It was observed in $M_2$ and $M_3$ that gamma rays showed higher mutagenic efficacy and generated good inherited variance for the yield-related traits. $M_4$ established three divergent groups in each blackgram cultivar revealed by clustering analysis. The number of pods per plant, number of clusters per plant, and number of pods per cluster showed a strong direct association with single plant yield and could be considered as selection traits. G × E interactions were higher than the variation due to genotype for single plant yield. Limited environmental interaction was observed for the genotypes G24, G16, G36, G30, and G17, as revealed by AMMI, and the genotypes G18 and G29, as revealed by GGE. GGE biplot revealed the environment-specific genotypes G13 for $E_1$ (Aduthurai), G7 for $E_2$ (Kattuthottam), and G34 for $E_3$ (Vamban) and also portrayed the highly discriminating ($E_3$) and representative ($E_2$) environments. Selected novel blackgram genotypes from this research are useful genetic stocks for genetic improvement and breeding.

**Keywords:** AMMI; G × E interaction; mutation breeding; pulses; blackgram

## 1. Introduction

Urdbean, also known as blackgram (*Vigna mungo*), is a nutritious and most commonly tailored stress-tolerant legume. It is a cheap source of vegetable protein, amino acids, etc. for Asian and African countries. Blackgram has been grouped mainly based on seed character and days to maturity into two primary categories, namely var. mungo and var. viridis; the former is characterized by large black seed and early maturity, and the latter is characterized by small greenish seed and late maturity. The black seeds (var. mungo) are predominant in the market. The crop plays a major role in improving soil fertility. It is also well suited for various cropping systems (i.e., dry farming and intercropping) [1]. In terms of healthy human nutrition, high lysine values make blackgram an ideal companion to rice. Blackgram originated in India and is predominantly grown in Asian countries such as India, Myanmar, Pakistan, Bangladesh, and Thailand. India is the leading blackgram producer and produces about 70% of world production [2]. It was cultivated on around

5.44 million hectares and resulted in 3.56 million tonnes of production in 2017–2018 [3]. In India and Thailand, it has been recorded that the mean seed yield of blackgram is low, with an average of 650–800 kg/ha [4]. Thus, the primary goal of any blackgram program is to improve the yield and its associated traits.

Successful breeding for yield-related traits requires genetic variation. The degree of existing natural genetic variation in the germplasm pool determines the traditional crop breeding program's accomplishment. However, owing to the autogamous flowering pattern and narrowed genetic polymorphism in the elite gene pool, achieving a genetic gain in blackgram through recombination is difficult. The small genome size (574 Mb) [5] and limited blackgram gene pools contribute to the weak basis of parent materials that have hindered blackgram breeding programs in recent years. Expanding genetic variation may offer better traits for the genetic improvement of the crop for sustainable food production and other qualities. Under these circumstances, mutation breeding provides scope for exploiting novel variants for yield-related traits in blackgram. Induced mutagenesis using radiation or chemical mutagens [6] creates a new allelic permutation in the traits of interest for genetic enhancement with no disruption in the plant's basic chromosome structure [7]. It relatively shortens the breeding cycle, not like spontaneous mutation and controlled recombination [8,9]. Mutagenesis was used to effectively tailor many plant characteristics, namely plant height, days to maturity, and pest and disease tolerance, in various legume crops, including mungbean [10], blackgram [11], cowpea [12,13], and lentil [7]. The first blackgram mutant variety in India, Co 4, was developed in 1978, and other blackgram mutant varieties include DU-1, Manikya, TAU-1, TAU-2, TPU-4, TAU-94-2, and Vamban 2. A combination of small (1–16 bp) and large (up to 130 kbp) deletion mutations have been produced in Arabidopsis and rice genomes using gamma irradiation [14,15]. Assessing genetic variability present in germplasm using morphological traits by multivariate analysis and understanding the associations between seed yield and yield-related traits would facilitate the selection of progenies through breeding cycles. This was studied earlier in blackgram [16,17] and soybean [18,19].

Identifying stable genotypes with the minimum environmental impact in terms of yield efficiency is more crucial [20]. More than a few statistical tools, for instance, additive main effects and multiplicative interaction (AMMI) model and the genotype main effects and genotype × environment interaction effects (GGE) model, are used widely to analyze and interpret G × E statistics [21]. Indeed, the GGE biplot tool is designed to compare genotype reactions across diverse locations and validate the test environment [20,21]. Revanappa et al. [22] studied G × E interaction for 11 blackgram genotypes and identified two stable genotypes (K-7-7 and DU-1) for grain yield. In another study, Konda et al. [17] evaluated 40 genotypes at three different locations for two seasons and identified four stable genotypes (TAU 1, 723, BDU 2, and BDU 4) for grain yield. Rita et al. [23] studied G × E interaction for 14 genotypes for yield and its component traits. G × E interactions for yield traits in rice using mutant populations were previously reported by Poli et al. [24] and Oladosu et al. [25]; such interactions for seed yield in chickpea were reported by Atta et al. [26].

In the present study, we developed the mutant populations ($M_1$–$M_5$) in the background of blackgram cultivars, namely ADT 3, Co 6, and TU 17-9. A total of 12,000 $M_2$ mutants were produced. The mutant genotypes were characterized and selected from $M_3$ to $M_5$. As a result, 36 promising mutant genotypes were identified. These mutants and their parents were used to study the G × E interactions for grain yield and ascertain stable mutants across different environmental conditions. To our knowledge, this is the first report that details the G × E interactions in blackgram mutant genotypes.

## 2. Materials and Methods

### 2.1. Plant Genetic Materials and Gamma Irradiation

The genetic variability was induced in blackgram cultivars ADT 3, Co 6, and TU 17-9 using different doses of physical mutagen (gamma rays). These cultivars are popular

among the farmers and highly recommended for cultivation in southern India's different agro-climatic zone. The seeds of the cultivars were collected from Tamil Nadu Rice Research Institute, Aduthurai, India; Department of Pulses, Tamil Nadu Agricultural University, Coimbatore, India; and National Pulses Research Centre, India. The viable seeds of three cultivars were gamma-irradiated with 200 gy (A1, C1, T1), 300 gy (A2, C2, T2), 400 gy (A3, C3, T3), and 500 gy (A4, C4, T4) with a Cobalt-60 ($^{60}$Co) radioisotope, sourced at the Centre for Plant Breeding and Genetics, Tamil Nadu Agricultural University, Tamil Nadu, India. Initially, based on lethal dose 50 ($LD_{50}$) values of germination and survival test, gamma irradiation concentration was determined during Rabi 2011-12.

### 2.2. Location, Experimental Design, and Development of Mutants

The selection was done in sequences of experiments carried out in three locations, namely National Pulses Research Centre (NPRC), Vamban (78°90 E 10°36 N with 93 above MSL); Tamil Nadu Rice Research Institute (TRRI), Aduthurai (79° E and 10.45° N and 19.5 m above MSL); and Soil and Water Management Research Institute (SWMRI), Kattuthottam (10°45′ N and 79° E with 50 m above MSL). The $M_1$ and $M_2$ generation mutants were raised at NPRC Vamban and TRRI Aduthurai during Kharif 2012 and Rabi 2012, respectively (Figure 1). $M_3$ and $M_4$ generation's mutants were developed at NPRC Vamban during Kharif 2013 and Rabi 2013, respectively, and $M_5$ generation mutants were evaluated during Rabi 2014. $M_1$, $M_2$, and $M_3$ generations' mutants were segregating progenies raised in the nonreplicated trial. $M_4$ generation mutants were assessed by adopting randomized block design (RBD) and replicated twice. $M_5$ generation mutants were evaluated at three different locations, namely TRRI Aduthurai, NPRC Vamban, and SWMRI Kattuthottam by adopting RBD with two replications. Details of weather parameters during crop growth stages are given in Supplementary Table S1. Mutant plants ($M_1$ to $M_5$) were planted in the seed-to-seed and row-to-row spacing of 20 cm and 30 cm, respectively. Cultivation and plant protection practices were followed according to TNAU-CPGA [27] to ensure healthy crop growth.

### 2.3. Selection Method, Data Collection, and Statistical Analysis

#### 2.3.1. Selection Method

The 300 irradiated seeds ($M_1$) from each treatment, namely A1, A2, A3, A4, C1, C2, C3, C4, T1, T2, T3, and T4, were grown along with respective parent controls. Seeds were harvested separately from the fertile $M_1$ plants, and 20 $M_2$ healthy seeds from 10 randomly selected $M_1$ plants were raised in the plant to progeny row for developing $M_2$ generation. The randomly selected 1200 $M_2$ individual plants (10 lines from each treatment of each variety) were raised as a plant to progeny row for generating $M_3$. The selection was imposed on $M_3$ mutants following the standard descriptor IBPGR [28]. The selection was made across the three varietal mutant populations, searching for mutants for earliness, increased number of pods per plant, increased pod length, increased pod numbers per cluster, increased seed numbers per pod, and single plant yield. Indeed, these traits were equally considered for the selection of the mutants across three varietal mutant populations. The selection was continued from $M_3$ to $M_5$ generation. The selected 543 $M_3$ mutant lines showing improved yield-related traits and their parental lines were grown to raise $M_4$ generations as single-plant progenies with two replications. A set of 36 uniform, nonsegregating mutant progenies showing desired traits were selected and bulked to generate $M_5$ for further evaluation.

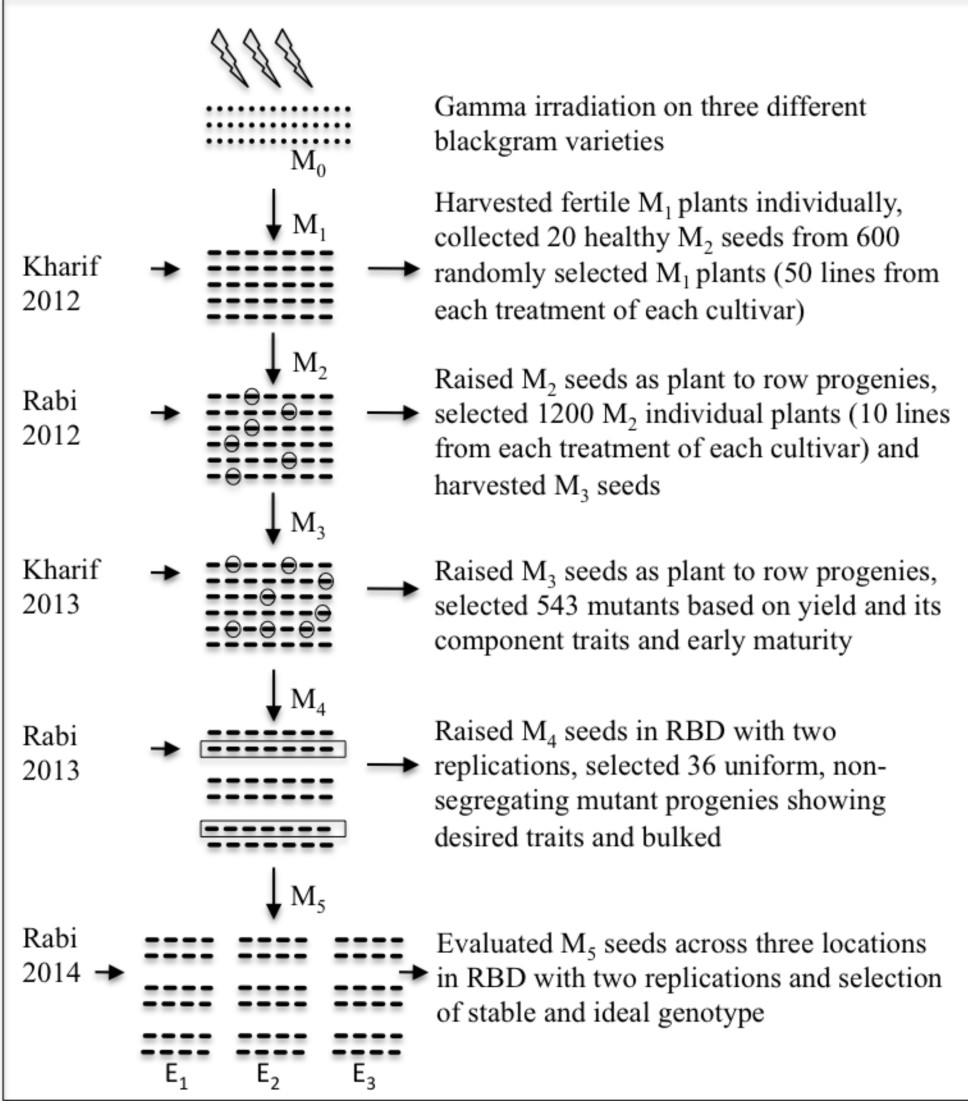

**Figure 1.** Schematic illustration for development of novel blackgram mutant and genotype × environment interaction for yield-related traits through mutation breeding.

### 2.3.2. Data Collection

The yield-related trait data, namely days to 50% flowering (DF), plant height (PH), number of basal branches (NB), number of clusters per plant (NC), number of pods per cluster (NPC), total number of pods per plant (NPP), pod length (PL), number of seeds per pod (NS) and single plant yield (SPY), were collected during evaluation throughout the selection cycle from $M_2$ to $M_4$. Data were recorded based on descriptors [28].

### 2.3.3. Statistical Analysis

Descriptive statistics associated with eight yield-related traits except DF observed in the current study across all the mutagenic treatments were calculated from 50 plant progeny of each treatment of $M_2$ and $M_3$ generation using Past 4 software [29]. In $M_4$, mean values of nine yield-related traits were used for descriptive statistics. Hierarchical squared Euclidean distance matrix-based cluster analysis assesses the degree of divergence and extent of heterogeneity created by different treatments of gamma irradiation. The percent contribution of yield-related traits to total genetic variation was estimated using principal component analysis (PCA). Correlation coefficients were employed to test the linear relationship among yield–related traits. These analyses were calculated with Past 4 software [29].

*2.4. $M_5$ Evaluation, GEI, and Stability of Mutant Lines for Grain Yield*

About 36 M5 generation mutant suitable lines selected for yield-related traits with their parents were evaluated across three environments. Three environments, namely experimental farm, TRRI, Aduthutai ($E_1$); experimental farm, Soil and Water Management Research Institute (SWMRI), Kattuthottam ($E_2$); and experimental farm, NPRC ($E_3$), were used to study the grain yield, adaptation, and breeding progress of mutant lines. The three parent varieties are the dominant blackgram cultivars in South Indian states; hence, they were considered as check varieties for this study. Each genotype across the environments was raised in a plot size of 1.2 m × 2 m in 5 rows of 2 m length, at approximately 40 plants per plot, and replicated twice. Five plants at random were taken from each plot in each replication under each environment to record the nine yield-related traits. Biometric observations were recorded using the methodology employed by Poli et al. [24] in rice and Rita et al. [23] in blackgram. As per the model described by Zobel et al. [30] and Crossa [31], the AMMI analysis was employed to evaluate associations between genotypes, environment, and G × E interaction. According to Yan and Tinker [32], data were analyzed graphically to interpret the relationship between G × E and define the stable and adaptive genotypes through the GGE path. Based on biplots, environmental vectors and the relationship between the environments were generated as described by Yan et al. [21]. GEA-R statistical software [33] was used for AMMI and GGE analysis.

## 3. Results

*3.1. Assessing the Effects of Mutagen on Yield-Related Traits in $M_1$ and $M_2$ Populations*

In $M_1$, 300 irradiated seeds from each treatment were raised ($M_1$). The results exhibited that the mean for yield-related traits such as NB, NC, NPP, NS, and SPY reduced drastically and showed a gamma ray dose dependent negative linear regression, except NB in ADT 3 and Co 6. The variety Co 6 was more sensitive than two other varieties to gamma rays (Data has not shown). About 20 healthy $M_2$ seeds from 50 randomly selected $M_1$ plants in each treatment of each cultivar (600 mutant lines in total) were harvested individually and used to raise the $M_2$ population consisting of approximately 12,000 mutant plants. The PH, NB, NPC, PL, and SPY trait means were reduced significantly from the parental populations and exhibited a negative shift from the control mean against each mutagenic treatment in all the three cultivars in $M_2$. However, a shift in mean occurred in both positive and negative directions in all the cultivars for the traits NS, NC (except cv. Co 6), and NPP (except A5). PH mean ranged from 3 (A1) to 65 cm (A4), 2.68 (C1) to 45.20 cm (C3), and 12 (T1) to 61.20 cm (T3). The mean of NB ranged between 1.0 (A1, A2, A3, A4) and 6.0 (A4), 0.0 (C4) and 8.0 (C2), and 0.0 (T3) to 6.0 (T3). The mean of NPP ranged between 10.0 (A2, A4) and 102.0 (A4), 4.0 (C2) and 102.0 (C3), and 4.0 (T4) and 92.0 (T3). NS ranged between 3.0 (A2) and 10.0 (A3, A4). SPY ranged between 1.84 (A4) and 19.61 g (A4), 1.08 (C3) and 19.57 g (C3), and 0.43 (T2) and 20.61g (T3) in cv. ADT 3, Co 6, and TU 17-9 mutant populations, respectively. The variation observed in leaf and seed characteristics is shown in Figure 2. The traits PH, NB, and NPC showed medium to high heritability (broad sense) estimates. Similarly, low to high heritability estimates were found for NC, PL, and NS. Conversely, all the treatments showed high heritability for the trait SPY (Table S2). Genetic variants derived through gamma irradiation from respective parents for yield-related traits were isolated and are presented in Figure 3.

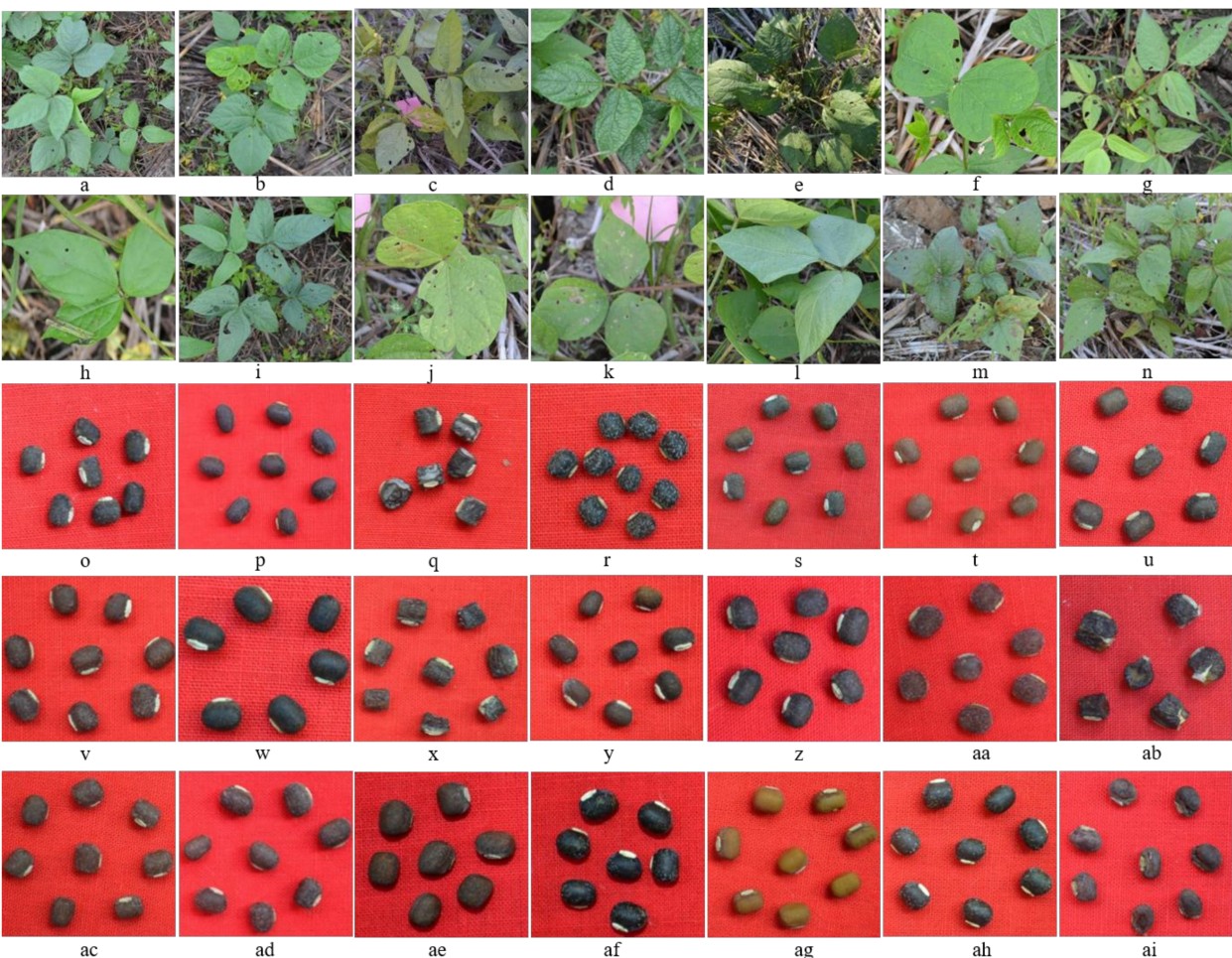

**Figure 2.** Variation observed in leaf and seed characteristics of cv. ADT 3, Co 6, and TU 17-9 mutagenized populations. (**a**) cv. ADT 3 (wild); (**b–g**) leaf variations in cv. ADT 3 mutagenized population. (**h**) cv. Co 6 (wild); (**i,j**) leaf variation in cv. Co 6 mutagenized population. (**k**) cv. TU 17-9 (wild); (**l–m**) leaf variations in cv. TU 17-9 mutagenized population. (**o**) cv. ADT 3 (wild); (**p–u**) seed characteristic variations in cv. ADT 3 mutagenized population. (**v**) cv. Co 6 (wild); (**w–ab**) seed characteristic variations in cv. Co 6 mutagenized population. (**ac**) cv. TU 17-9 (wild); (**ad–ai**) seed characteristic variations in cv. TU 17-9 mutagenized population.

### 3.2. Assessing Genetic Heterogeneity on Yield-Related Traits in M₃ Population

The $M_3$ population was raised with approximately 12,000 mutant plants, consisting of about 10 healthy $M_3$ seeds from randomly selected 10 $M_2$ lines from each cultivar's treatment (1200 mutant lines in total). The mean values of PH, NB, and PL for different gamma treatments shifted towards the negative sign for the cv. ADT 3 compared to its parent and ranged between 12.0 (A1) and 42.0 cm (A1, A2), 0.0 (A3, A4) and 6.0 cm (A1), and 2.40 (A3) and 6.40 cm (A1), respectively (Table S3). Correspondingly, NC, NPP, and SPY varied from 3.0 to 21.0 g, 5.0 to 53.0 g, and 0.06 to 8.07 g. Conversely, the traits NPC and NS recorded a positive shift compared to the control. Treatment means of cv. Co 6 across treatments were increased for the traits SPY and PL. SPY ranged between 0.99 (C2) and 17.03 g (C1). Values for traits PH, NB, NC, NPC, NPP and NS were in the ranges of 12.0 (C3) to 60 (C1), 0.0 (C1) to 8.0 (C1), 3.0 (C3, C4) to 30.0 (C2), 2.0 to 4.0, 5.0 (C4) to 67 (C1), and 3.0 (C2) to 9.0 (C3, C4), respectively. Concerning cv. TU 17-9, the treatment means were found to be increased for the traits PH and PL. SPY ranged from 0.08 to 10.56. The majority of the treatments of all three cultivars exhibited high heritability. However, C3 and C4 for PH, A3 and A4 for NB, A2 and A3 for NC, and A2 and C3 for NPC recorded moderate

heritability estimates. The selection was imposed on $M_3$ mutants, and 543 mutants were selected based on yield-related traits to raise the $M_4$ population.

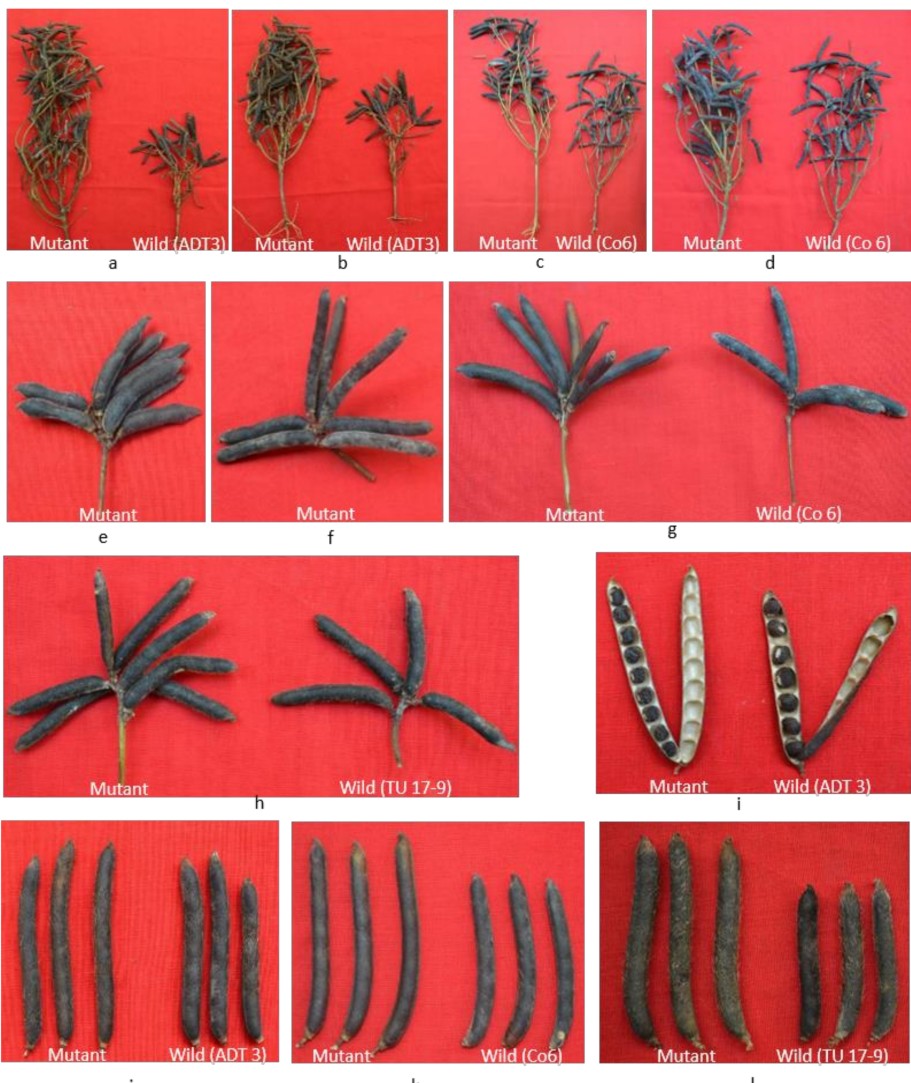

**Figure 3.** Mutants showing variations in yield-related traits: (**a,b**) mutants derived from cv. ADT 3 exhibiting increased pod numbers per plant, (**c,d**) mutants derived from cv. Co 6 showing increased pod numbers per plant; (**e–g**) mutants derived from cv. Co 6 exhibiting increased pod numbers per cluster; (**h**) mutant derived from cv. TU 17-9 showing increased pod numbers per cluster; (**i**) mutant derived from cv. ADT 3 exhibiting increased seed numbers per pod; (**j–l**) mutant derived from cv. ADT 3, Co 6, and TU 17-9 showing increased pod length.

### 3.3. Assessing Genetic Parameters on Yield-Related Traits in the $M_4$ Population

$M_4$ was established from the selected 543 $M_3$ mutants in the replicated trial, and 124 nonsegregating mutant lines were considered for recording biometric observations. Analysis of variance (ANOVA) for DF, PH, NB, NC, NPC, NPP, PL, NS, and SPY was significant, indicating a significant difference among the mutant population (Table 1). The traits NC (r = 0.648 **, 0.813 **, 0.767 **), NPC (r = 0.682 **, 0.530 **, 0.740 **), and NPP (r = 0.754 **, 0.818 **, 0.851 **) had strong association with grain yield for the cv. ADT 3, Co 6, and TU 17-9, respectively (Table 2). Principal components (PCs) defined the percent variation explained by yield-related traits, as given in Table 3. The PC1 was responsible for 39.62% and 39.88% of the total variation, and PC1 separates accessions on five traits, namely PH, NB, NC, PL, and SPY; the PC2 accounted for 18.97% and

18.20% of the total variation, mainly attributed to NPC and NPP of cv. ADT 3 and cv. Co 6, respectively. When considering TU 17-9 mutant genotypes, PC1, PC2, and PC3 accounted for 46.85, 19.10, and 10.08% of the variation and were largely contributed to NC, PL, and PH; NPP and NPC; and NS and SPY of PC1, PC2, and PC3, respectively. The eigenvalues for PC1, PC2, and PC3 were 3.57, 1.71, and 1.14 for cv. ADT 3; 3.59, 1.64, and 1.22 for cv. Co 6; and 4.22, 1.72, and 0.91 for cv. TU 17-9, respectively. The mutant populations of each blackgram cultivar formed three clusters; for cv. ADT 3, cluster I consisted of 8 genotypes, cluster II formed a larger size with 37 genotypes, and cluster III comprised 15 genotypes. Similarly, for cv. Co 6, clusters I, II, and III comprised 20, 3 and 16 genotypes, respectively. For cv. TU 17-9, clusters I, II, and III comprised 8, 17, and 3 genotypes, respectively (Figure 4). The selection was imposed on $M_4$ mutants, and 36 uniform, nonsegregating mutant progenies showing desired yield-related traits were selected to raise the $M_5$ population.

**Table 1.** Mean and range for nine yield-related traits of $M_4$ mutant populations of three blackgram cultivars.

| Parameter | cv. ADT 3 | | | cv. Co 6 | | | cv. TU 17-9 | | |
|---|---|---|---|---|---|---|---|---|---|
| | Mean | Min | Max | Mean | Min | Max | Mean | Min | Max |
| DF | 38.93 | 36.00 | 41.50 | 39.91 | 38.00 | 41.00 | 37.56 | 35.00 | 41.50 |
| PH | 22.37 | 14.75 | 32.84 | 26.86 | 18.00 | 36.50 | 22.12 | 14.25 | 32.00 |
| NB | 2.85 | 1.25 | 4.85 | 2.82 | 1.65 | 4.50 | 2.95 | 1.50 | 5.00 |
| NC | 11.88 | 6.00 | 22.25 | 13.31 | 4.75 | 24.25 | 12.06 | 5.50 | 20.25 |
| PC | 2.86 | 2.00 | 4.00 | 3.11 | 2.00 | 4.00 | 2.84 | 2.00 | 3.85 |
| NPP | 26.62 | 8.75 | 45.75 | 33.45 | 12.25 | 57.90 | 26.36 | 13.00 | 51.25 |
| PL | 4.89 | 4.03 | 5.79 | 4.85 | 4.00 | 5.45 | 4.78 | 4.09 | 5.35 |
| NS | 6.56 | 5.50 | 7.75 | 6.31 | 5.75 | 7.00 | 6.38 | 5.00 | 7.00 |
| SPY | 4.43 | 0.64 | 8.81 | 4.52 | 1.39 | 9.02 | 4.30 | 1.51 | 9.23 |

**Table 2.** Character association for yield-related traits in mutant populations of three blackgram cultivars.

| Quantitative Traits | cv. ADT 3 | cv. Co 6 | cv. TU 17-9 |
|---|---|---|---|
| | Grain Yield (g) | Grain Yield (g) | Grain Yield (g) |
| Days to 50% flowering | −0.013 | 0.181 | 0.443 * |
| Plant height (cm) | 0.537 ** | 0.187 | 0.493 ** |
| Number of fertile branches | 0.234 | 0.564 ** | 0.513 ** |
| Number of clusters per plant | 0.648 ** | 0.813 ** | 0.767 ** |
| Number of pods per cluster | 0.682 ** | 0.530 ** | 0.740 ** |
| Number of pods per plant | 0.754 ** | 0.818 ** | 0.851 ** |
| Pod length (cm) | 0.096 | 0.084 | 0.197 |
| Number of seeds per pod | −0.174 | 0.085 | 0.141 |

* and ** indicates significance at 5% and 1% level.

**Table 3.** Vector loadings and percentage of variation explained by the first three principal components after assessing the yield-related traits in a mutant population of three blackgram cultivars.

| Traits | cv. ADT 3 Mutants | | | cv. Co 6 Mutants | | | cv. TU 17-9 Mutants | | |
|---|---|---|---|---|---|---|---|---|---|
| | PC 1 | PC 2 | PC 3 | PC 1 | PC 2 | PC 3 | PC 1 | PC 2 | PC 3 |
| Days to 50% flowering | 0.379 | −0.154 | 0.768 | 0.607 | −0.320 | −0.439 | 0.343 | −0.092 | 0.335 |
| Plant height (cm) | 0.879 | 0.047 | 0.230 | 0.934 | −0.095 | −0.170 | 0.444 | −0.078 | −0.024 |
| Number of branches per plant | 0.839 | 0.003 | −0.229 | 0.640 | 0.104 | 0.530 | 0.363 | −0.106 | −0.518 |
| Number of clusters per plant | 0.934 | −0.007 | 0.081 | 0.878 | −0.199 | −0.035 | 0.457 | −0.153 | −0.161 |
| Number of pods per cluster | −0.037 | 0.904 | −0.072 | 0.261 | 0.780 | −0.213 | 0.060 | 0.640 | −0.243 |
| Number of pods per plant | −0.127 | 0.648 | 0.534 | 0.235 | 0.782 | −0.144 | −0.010 | 0.681 | −0.068 |
| Pod length (cm) | 0.872 | 0.109 | −0.194 | 0.896 | −0.229 | 0.008 | 0.446 | 0.106 | −0.180 |
| Number of seeds per pod | −0.123 | 0.619 | −0.135 | 0.186 | 0.012 | 0.798 | 0.254 | 0.171 | 0.616 |
| Single plant yield (g) | 0.528 | 0.223 | −0.301 | 0.456 | 0.453 | 0.114 | 0.277 | 0.191 | 0.342 |
| Eigenvalue | 3.57 | 1.71 | 1.14 | 3.59 | 1.64 | 1.22 | 4.22 | 1.72 | 0.91 |
| % variance | 39.62 | 18.97 | 12.65 | 39.88 | 18.20 | 13.55 | 46.85 | 19.10 | 10.08 |
| % cumulative of variance | 39.62 | 58.59 | 71.24 | 39.88 | 58.08 | 71.63 | 46.85 | 65.95 | 76.03 |

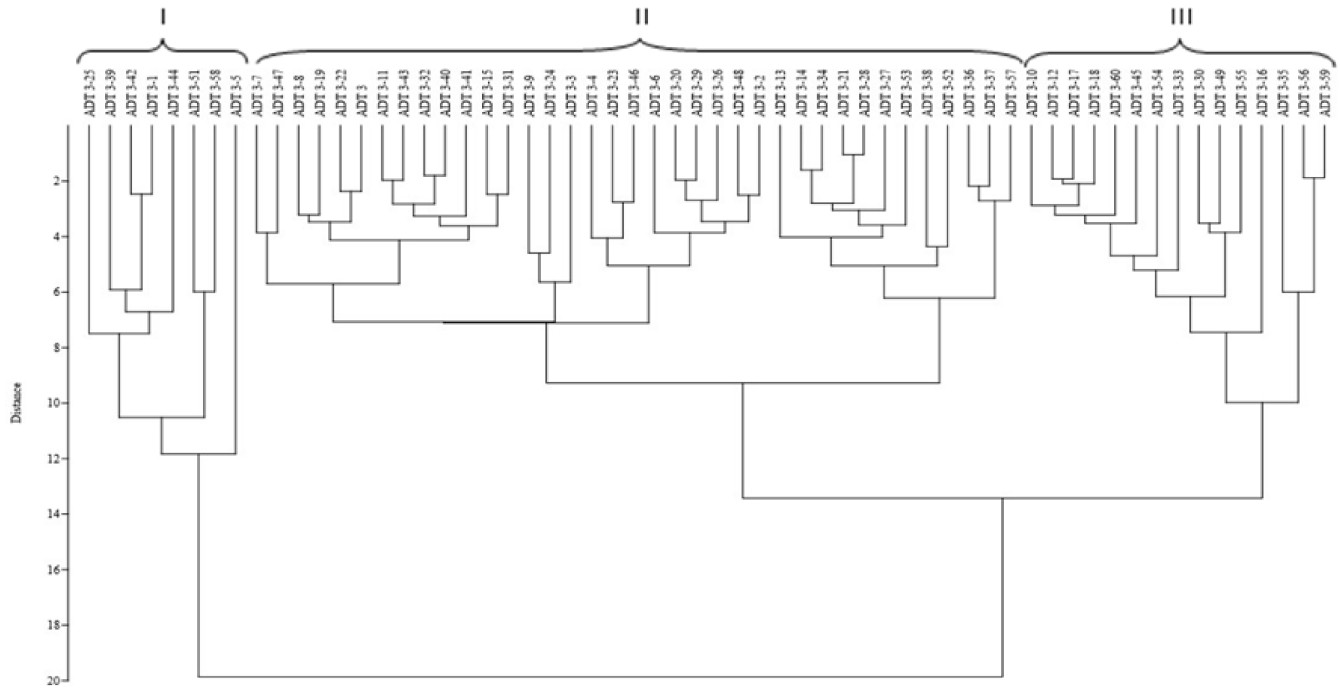

(**a**)

**Figure 4.** *Cont.*

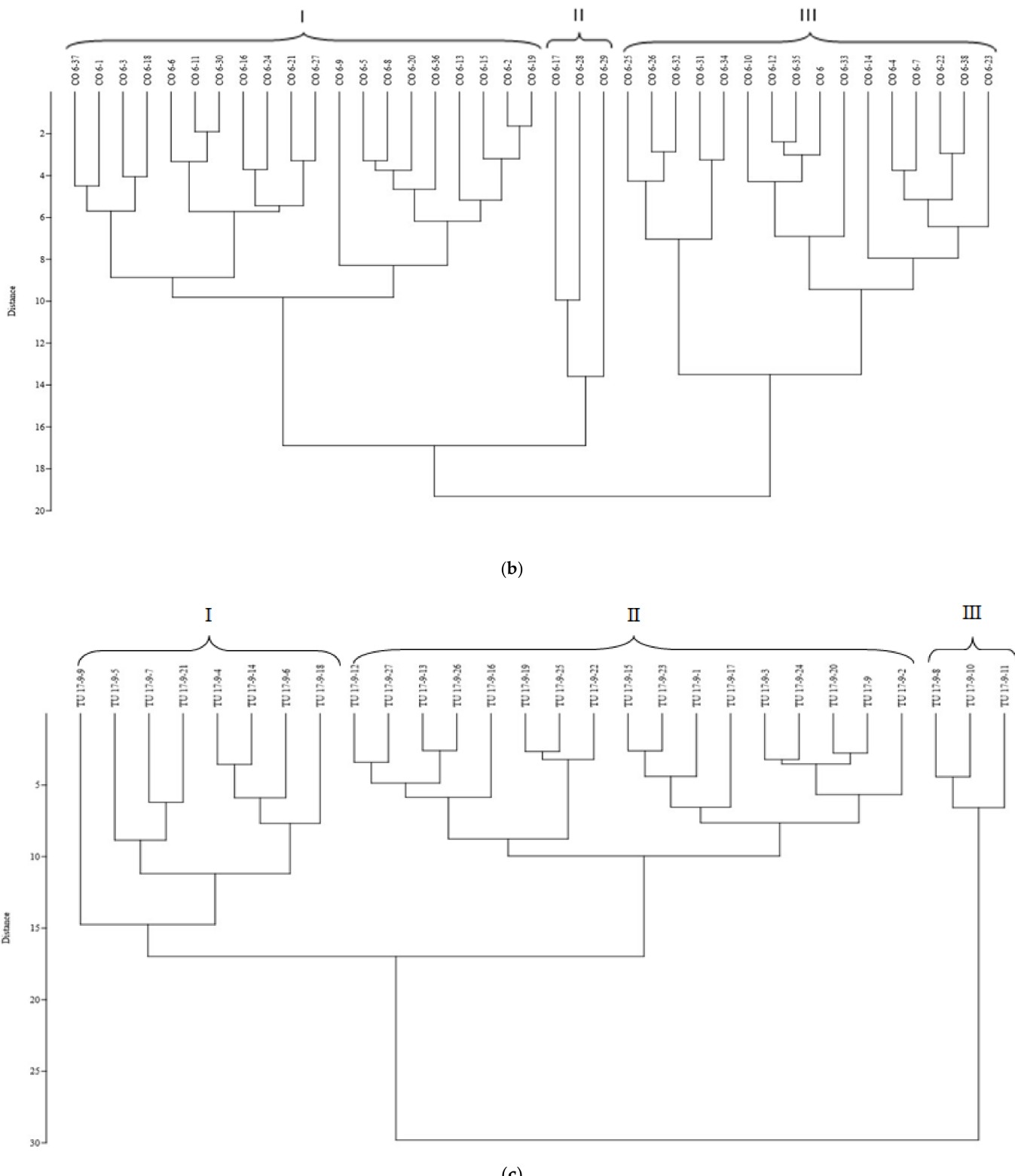

**Figure 4.** Dendrograms of blackgram mutant population clustering based on nine yield-related traits: (**a**) dendrogram using Euclidean similarity index and UPGMA algorithm for cv. ADT 3 mutant population; (**b**) dendrogram using Euclidean similarity index and UPGMA algorithm for cv. Co 6 mutant population; (**c**) dendrogram using Euclidean similarity index and UPGMA algorithm for cv. TU 17-9 mutant population.

### 3.4. Assessing GEI and Stability of Mutant Lines for Grain Yield in $M_5$ Population

A set of selected 36 $M_4$ mutants and three parents in a replicated trial across three environments were sown to raise the M5 population.

#### 3.4.1. ANOVA and per se Grain Yield Performance

The AMMI analysis of variance revealed that 34.86% of the sum of squares was attributed to genotype, 23.58% to the environment, and 41.56% to G × E interaction (GEI) effects (Table 4). The multiplicative variance for treatment sums of squares was divided into two major interaction components. The contribution of IPCA-1 was 57.24% and that of IPCA-2 was 42.76%. The AMMI-I and AMMI-II biplots were established to demonstrate the genotype and environmental effects concurrently (Figure 5a,b). The mean grain yield of mutant genotypes across three locations ranged from 3.46 (G9) to 6.3 g (G13) (Table 5). The gain in selecting a mutant over its respective parent/check was calculated based on mean values of the trait single plant yield across three environments, and the results are presented in Table 5. The gain in selection ranged from −35 to 73%.

**Table 4.** AMMI analysis of variance for 36 blackgram mutants and their three parents across three different environments.

| | Sum of Squares | Degrees of Freedom | Mean sum of Squares | *F*-Value | Probability | Variation Explained (%) |
|---|---|---|---|---|---|---|
| Environment | 107.55 | 2.00 | 53.77 | 29.98 | 0.00 | 23.58 |
| Genotype | 159.02 | 38.00 | 4.18 | 2.33 | 0.00 | 34.86 |
| Environment × Genotype | 189.57 | 76.00 | 2.49 | 1.39 | 0.05 | 41.56 |
| PC1 | 108.52 | 39.00 | 2.78 | 1.55 | 0.04 | 57.24 |
| PC2 | 81.05 | 37.00 | 2.19 | 1.22 | 0.21 | 42.76 |
| Residuals | 209.85 | 117.00 | 1.79 | | | |

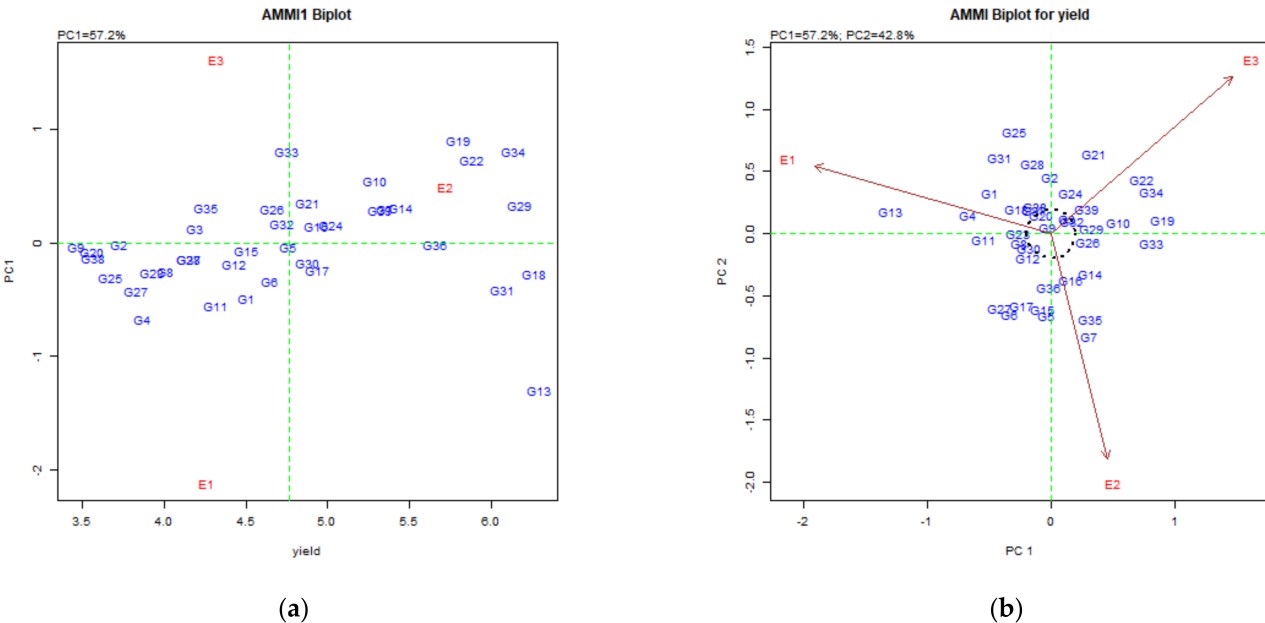

(**a**)                                        (**b**)

**Figure 5.** AMMI model: (**a**) AMMI-I explaining main and IPCA-1 interaction for grain yield; (**b**) AMMI-II explaining interaction effect of IPCA-1 and IPCA-2 for grain yield.

Table 5. Mean single plant yield, principal component analysis of blackgram mutant accessions, parental genotypes, and gain in selection (%) over respective parent/check.

| S.No | Codes | Source | Aduthurai (E$_1$) | Kattuthottam (E$_2$) | Vamban (E$_3$) | Mean | Gain in Selection (%) | PC 1 | PC 2 |
|---|---|---|---|---|---|---|---|---|---|
| 1 | G1 | ADT 3 derived mutant | 5.23 | 4.56 | 3.71 | 4.50 | 8 | −0.38 | 0.25 |
| 2 | G2 | ADT 3 derived mutant | 3.52 | 3.77 | 3.88 | 3.72 | −10 | −0.01 | 0.35 |
| 3 | G3 | ADT 3 derived mutant | 3.49 | 4.97 | 4.10 | 4.19 | 1 | 0.09 | 0.09 |
| 4 | G4 | TU 17-9 derived mutant | 4.88 | 4.19 | 2.52 | 3.86 | −27 | −0.52 | 0.11 |
| 5 | G5 | Co 6 derived mutant | 3.94 | 7.02 | 3.32 | 4.76 | 33 | −0.03 | −0.51 |
| 6 | G6 | Co 6 derived mutant | 4.45 | 6.75 | 2.74 | 4.65 | 30 | −0.26 | −0.50 |
| 7 | G7 | ADT 3 derived mutant | 3.70 | 8.12 | 4.22 | 5.35 | 29 | 0.23 | −0.64 |
| 8 | G8 | Co 6 derived mutant | 3.99 | 4.99 | 3.04 | 4.01 | 12 | −0.20 | −0.06 |
| 9 | G9 | TU 17-9 derived mutant | 3.05 | 4.29 | 3.04 | 3.46 | −35 | −0.02 | 0.04 |
| 10 | G10 | ADT 3 derived mutant | 3.69 | 6.33 | 5.86 | 5.29 | 27 | 0.42 | 0.07 |
| 11 | G11 | ADT 3 derived mutant | 4.94 | 5.10 | 2.90 | 4.32 | 4 | −0.42 | −0.04 |
| 12 | G12 | ADT 3 derived mutant | 4.20 | 5.69 | 3.40 | 4.43 | 7 | −0.15 | −0.15 |
| 13 | G13 | ADT 3 derived mutant | 8.64 | 6.26 | 3.99 | 6.30 | 52 | −1.00 | 0.13 |
| 14 | G14 | ADT 3 derived mutant | 4.08 | 7.22 | 5.05 | 5.45 | 31 | 0.24 | −0.25 |
| 15 | G15 | ADT 3 derived mutant | 3.77 | 6.66 | 3.09 | 4.50 | 8 | −0.05 | −0.47 |
| 16 | G16 | ADT 3 derived mutant | 3.87 | 6.71 | 4.21 | 4.93 | 19 | 0.12 | −0.29 |
| 17 | G17 | ADT 3 derived mutant | 4.59 | 6.93 | 3.28 | 4.94 | 19 | −0.19 | −0.44 |
| 18 | G18 | ADT 3 derived mutant | 6.46 | 6.69 | 5.66 | 6.27 | 51 | −0.21 | 0.15 |
| 19 | G19 | ADT 3 derived mutant | 3.46 | 6.98 | 6.97 | 5.80 | 40 | 0.69 | 0.09 |
| 20 | G20 | ADT 3 derived mutant | 3.31 | 4.17 | 3.19 | 3.56 | −14 | −0.06 | 0.12 |
| 21 | G21 | ADT 3 derived mutant | 4.01 | 4.73 | 5.89 | 4.88 | 18 | 0.27 | 0.49 |
| 22 | G22 | ADT 3 derived mutant | 4.09 | 6.32 | 7.22 | 5.88 | 42 | 0.56 | 0.34 |
| 23 | G23 | ADT 3 derived mutant | 3.98 | 4.74 | 3.05 | 3.92 | −6 | −0.20 | 0.01 |
| 24 | G24 | ADT 3 derived mutant | 4.38 | 5.40 | 5.28 | 5.02 | 21 | 0.12 | 0.25 |

**Table 5.** *Cont.*

| S.No | Codes | Source | Aduthurai (E$_1$) | Kattuthottam (E$_2$) | Vamban (E$_3$) | Mean | Gain in Selection (%) | PC 1 | PC 2 |
|------|-------|--------|-------------------|----------------------|----------------|------|----------------------|------|------|
| 25 | G25 | TU 17-9 derived mutant | 4.29 | 2.83 | 3.88 | 3.67 | −31 | −0.23 | 0.63 |
| 26 | G26 | Co 6 derived mutant | 3.48 | 5.90 | 4.59 | 4.66 | 31 | 0.23 | −0.05 |
| 27 | G27 | Co 6 derived mutant | 3.85 | 5.78 | 1.86 | 3.83 | 7 | −0.32 | −0.46 |
| 28 | G28 | Co 6 derived mutant | 4.29 | 3.90 | 4.25 | 4.15 | 16 | −0.11 | 0.43 |
| 29 | G29 | Co 6 derived mutant | 5.00 | 7.22 | 6.32 | 6.18 | 73 | 0.25 | 0.03 |
| 30 | G30 | Co 6 derived mutant | 4.68 | 5.97 | 3.98 | 4.88 | 37 | −0.14 | −0.09 |
| 31 | G31 | TU 17-9 derived mutant | 6.81 | 5.58 | 5.81 | 6.07 | 14 | −0.32 | 0.47 |
| 32 | G32 | TU 17-9 derived mutant | 3.91 | 5.56 | 4.68 | 4.72 | −11 | 0.13 | 0.08 |
| 33 | G33 | TU 17-9 derived mutant | 2.49 | 6.25 | 5.50 | 4.75 | −11 | 0.62 | −0.06 |
| 34 | G34 | TU 17-9 derived mutant | 4.11 | 6.82 | 7.48 | 6.14 | 15 | 0.62 | 0.26 |
| 35 | G35 | ADT 3 derived mutant | 2.67 | 6.74 | 3.35 | 4.25 | 2 | 0.24 | −0.53 |
| 36 | G36 | ADT 3 derived mutant | 4.93 | 7.46 | 4.57 | 5.66 | 36 | −0.01 | −0.33 |
| | | Parents/checks | | | | | | | |
| 37 | G37 | ADT 3 | 4.05 | 4.66 | 3.73 | 4.15 | | −0.11 | 0.14 |
| 38 | G38 | Co 6 | 3.47 | 4.01 | 3.21 | 3.57 | | −0.10 | 0.17 |
| 39 | G39 | TU 17-9 | 4.32 | 6.02 | 5.62 | 5.32 | | 0.22 | 0.15 |
| | Mean | | 4.26 | 5.73 | 4.32 | 4.77 | | | |
| Range | | | 2.49–8.64 | 2.83–8.12 | 1.86–7.48 | | | | |

### 3.4.2. Identification of Genotypes for Favorable Environments

The mutant genotypes G24 and G16 were identified as specifically adapted to favorable environments as they exhibited a high main effect, showed positive interaction with $E_2$ and $E_3$, had a positive IPCA-1 score. On the other hand, $E_1$ was found to be a favorable environment for mutant genotypes G36, G30, and G17 as it recorded a negative IPCA-1 score for genotypes and environment with high main effect.

### 3.4.3. Identification of Genotypes for Broad-Spectrum Adaptability

The mutant genotypes G24, G16, G36, G30, and G17 exhibited IPCA-1 scores nearer to zero with good yielding ability; they are stable and generally adapted to all the environments under study. Environments under investigation were discriminatory since they situated far from the biplot origin specified by the AMMI-II biplot. The mutant genotypes G3, G9, and G20 were nearer to the origin, suggesting limited interaction between these genotypes and the environment. The genotypes G13, G25, G21, G22, G34, G19, G33, G7, G6, and G27 were scattered away from the origin and were more susceptible to environmental interaction forces (positive or negative) (Table 5).

### 3.4.4. Discovering Promising Genotypes

The GGE biplot enables environmental assessment based on the GGE views of discriminative potential and representativeness. The association between testing environments has been studied from environment-centered (centering, 2) and environment-metric-preserving (SVP, 2) perspectives without a scaling choice. The which-won-where pattern analysis can assign the genotypes to every environment. The polygon is divided by rays (color lines) that begin from the biplot origin and pass the polygon sides vertically, separating it into seven sectors. $E_1$ is highly appropriate for the genotypes G13, G31, G18, G1, G17, and G30 as they are situated in the same region. In contrast, genotypes G29, G34, G19, G22, G10, G7, G14, G24, and G39 were highly desirable for $E_2$ and $E_3$ (Figure 6a). The best performing genotypes are plotted at polygon vertexes, and genotypes G13, G29, and G34 and G19 were placed nearer to the vertexes for the environments $E_1$, $E_2$, and $E_3$, respectively.

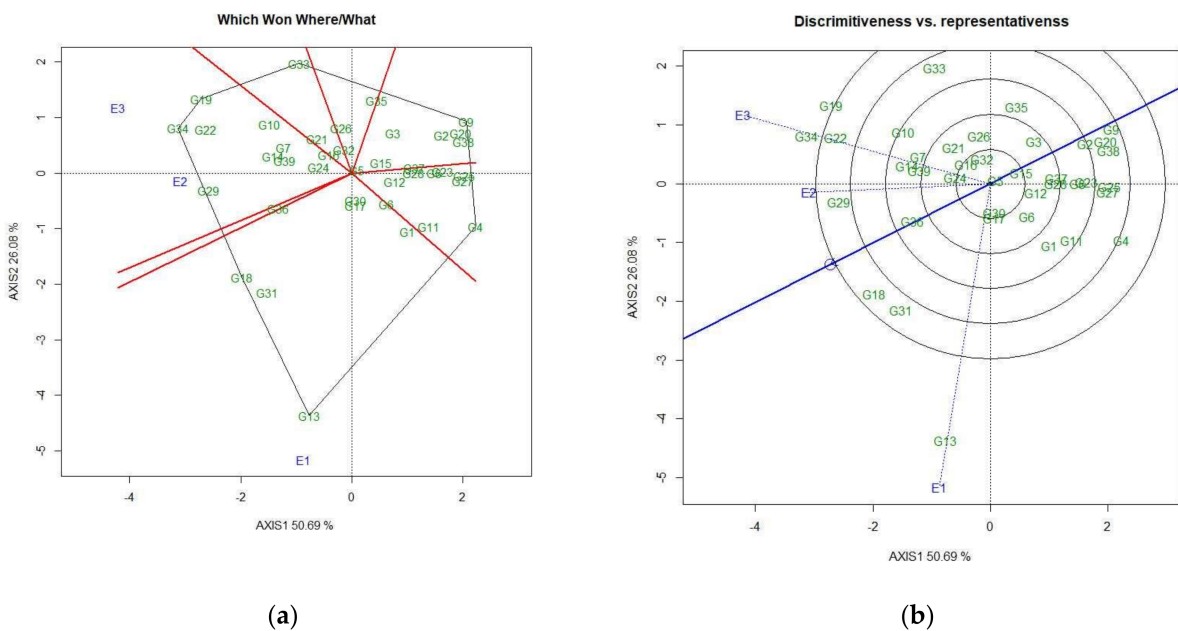

(**a**)  (**b**)

**Figure 6.** *Cont.*

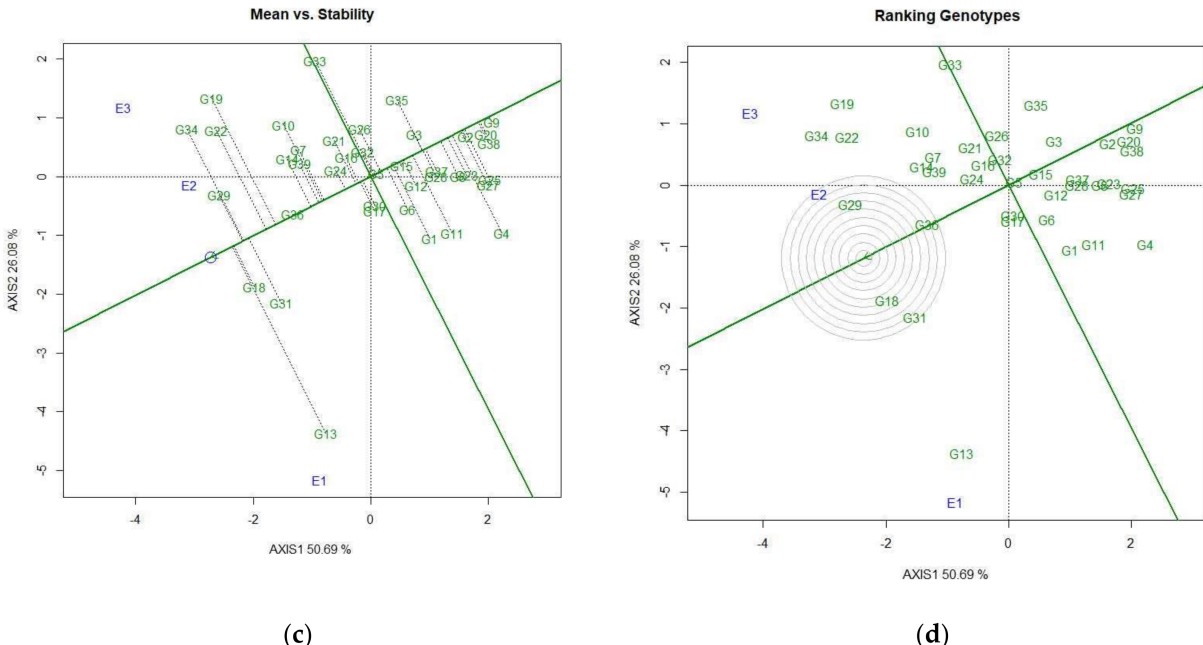

(**c**)　　　　　　　　　　　　　　　　　　　　　　(**d**)

**Figure 6.** GGE biplot view: (**a**) polygon view delineating successful genotypes in each environment; (**b**) discriminating and representative ability of the environment for grain yield; (**c**) AEC view showing the relationship between grain yield and stability for identification of stable genotypes; (**d**) discovering ideal genotypes by ranking genotypes.

The present research showed that $E_1$ was more discriminating and $E_2$ was a highly representative environment as they had the longest vector and smallest angle, respectively (Figure 6b), demonstrating their suitability as research environments for the multienvironment trial. The ideal environment can be identified using the average environment axis (AEA). The stability of the genotypes can be predicted using the average environment coordinate (AEC). A stable genotype displays a short upright line to the AEC axis. In contrast, the increase in perpendicular line length shows a decline in genotype stability. Genotypes G36, G5, G15, G2, and G9 were very stable and showed good to low grain yield potential. G13 was extremely unstable; along with G33, G19, and G34, it was very far off the AEC axis and expressed good to average yield capability (Figure 6c). The genotypes were ranked considering all three locations for grain yield (Figure 6d). The biplot described the stable genotypes with the longest vectors, along with zero $G \times E$, by points and arrows. It was seen that the AEA projection was near zero for G18 and G29 followed by G31 and G36.

## 4. Discussion

The degree of genetic variation in the crop gene pool influences the success of breeding. Irradiation is a proven technology for creating novel genetic combinations, including economic traits in the crop genome. Researchers considered that genetic polymorphism in the mutagenized population is due to induced micro-mutations. Reduction in mean for yield-related traits across three cultivars has been observed; concentration of gamma rays proportionally increases the pronounced impeding effects on yield-related traits increases. When considering the $M_1$ effects, cultivar Co 6 was found to be more sensitive to gamma rays than two other varieties. Similar results were obtained earlier in blackgram [34,35].

Mean values of yield-related traits recorded shifts in either direction in $M_2$. However, the inhibition of characteristic expression at higher gamma concentrations was maximal. The lesser or medium gamma concentrations were allied with an improved mean for yield-related traits. Observation of different yield-related traits in $M_2$ and $M_3$ revealed a wide range of trait variation created by gamma rays across cultivars because of polygenes' involvement in the expression of such yield-related traits and also because of random interaction of mutagen with the targeted genome. The increased mean of yield-related traits was more prominent in $M_3$ than in $M_2$ generation, given that $M_3$ mutants might

go through recombination twice and experience more effective removal of undesirable mutations than that occurring in $M_2$. The decline in the mean of a few traits in the $M_3$ generation was attributed to residual heterozygosity. Low, medium, and high heritability estimates were observed in $M_2$. The gain in heritability from $M_2$ to $M_3$ generation in the established mutant populations demonstrates the usefulness of the selected mutant lines. The current study corroborates the findings of Wani [36].

Results revealed that all radiation levels (doses)/treatments contributed significantly to the creation of mutants (Tables S4 and S5). In $M_2$, about 22, 32, 19, and 18 ADT 3 mutants showing distinct morphological variation were selected from A1, A2, A3, and A4, respectively. Similarly, for cv. Co 6, 16, 15, 19, and 12 mutants were selected from C1, C2, C3, and C4, respectively; for cv. TU 17-9, 12, 12, 14, and 10 mutants were selected from T1, T2, T3, and T4, respectively. Similarly, 543 and 36 mutants were selected for yield-related traits in $M_3$ and $M_4$.

From the biometric observations recorded in $M_4$, it was noted that the ANOVA and basic descriptive statistics revealed significant polymorphism among the nine yield-related traits of blackgram. NC, NPC, and NPP exhibited a significant and positive association with grain yield for the cultivars ADT 3, Co 6, and TU 17-9, demonstrating that selection through these traits can be used for identifying improved yield mutants [37,38]. Our result corroborated findings in rice bean [39], mungbean [40], and cowpea [41]. $M_4$ mutant populations of each cultivar formed three clusters. This suggested that gamma rays showed high mutagenic efficacy that leads to the creation of heterogeneous populations in the background of the three parental cultivars. Similar results were found by Senapati and Misra [42] and Kuralarasan et al. [43] in blackgram, Das and Baisakh [44] in mungbean, and Gnankambary et al. [45] in cowpea. The yield-related traits contribute reasonably to total genetic variation among the mutagenized population revealed by loading plot values in PCs. Therefore, subsequently, selecting these traits in the blackgram breeding program could lead to the selection of desirable lines. Afuape et al. [46] opined that the choice of suitable lines for further breeding could be successful through PCA. Similar finding was reported in lentils [7] and cowpea [41].

In this study, $G \times E$ interaction for the selected $M_5$ blackgram mutant genotypes for grain yield was studied across the three environments. Target environments were the major blackgram growing areas in the South Indian states; hence, these environments were selected for conducting $G \times E$ statistics for 36 mutants along with their respective parents, ADT 3, Co 6, and TU 17-9. These are the three dominant blackgram cultivars in South Indian states and were also considered as check varieties for this study and compared with the mutant genotypes. The efficacy of the mutagen varied between cultivars. Gamma rays at 400 and 500 gy were found to be more efficient in generating useful mutants. Gamma rays created a useful variant in which 14 $M_5$ mutants exhibited more than 20% gain in selecting the trait single plant yield. This result shows that when one-fourth of the sum of squares variation is attributed to the environment, environments are found to be sufficiently varied and cause more significant variation in grain yield [47,48]. A significant $G \times E$ interaction represents the adaptation of yield trait for a specific environment, and $G \times E$ interaction sum of squares was found to be 1.19 times higher than that of genotypes, demonstrating high genotype response variations across environments. Similar results were previously reported by Alam et al. [48], Tonk et al. [49], and Vaezi et al. [50]. From AMMI analysis, it was found that environments $E_2$ and $E_3$ were the favorable environments for G24 and G16, respectively, and $E_1$ was found to be the favorable environment for G36, G30, and G17, as these genotypes and their respective environments have the same sign of IPCA score. Further, environments E1 and E3 showed contrast in interaction patterns because of their differences in soil type, soil pH, altitude, rainfall, and maximum and minimum temperature during the study period. GGE analysis showed that $E_1$ was a more discriminating environment, and $E_2$ was found to be a highly representative environment as it had optimum soil pH and received good rainfall during flowering and pod filling stages as compared to $E_1$ and $E_3$; thus, $E_1$, $E_2$ and $E_3$ were demonstrated as suitable study

environments for multiple environmental trials. G36, G5, G15, G2, and G9 were stable and had good to low yield power. The which-won-where pattern analysis can assign the genotypes to each environment. The polygon is divided by rays (color lines) that begin from the biplot origin and pass the polygon sides vertically, separating it into seven sectors.

G13, G29, and G34 and G19 were the most suitable genotypes, being plotted near polygon vertexes for environments $E_1$, $E_2$, and $E_3$, respectively. $E_1$ and $E_2$ were well discriminating and representative environments, respectively, as revealed by AEA. A vertical line to the AEC axis indicated that G36, G5, G15, G2, and G9 were stable and good- to low-yielding genotypes. Similar results were reported for cowpea [51] and mungbean [50–53]. The projection on the AEA was nearer to zero for the genotypes G18 and G29 followed by G31 and G36, indicating the limited interaction between these genotypes and environments. The study discovered that G13, G7, and G34 could win in $E_1$, $E_2$, and $E_3$, respectively. These winning mutant genotypes can be used for developing location-specific blackgram cultivars. Similar findings were reported in durum wheat [54]. The identified stable genotypes suited to the specific and the general environment will be restudied statewide. These mutant genotypes may be utilized in future breeding programs and for understanding the genetic control of trait expression. In addition, the genotypes developed in this study could be useful for *Vigna* improvement programs in tropical countries.

## 5. Conclusions

The current study results indicate that gamma irradiation establishes the opportunities for the selection of elite mutants for improving yield and creates significant interpopulation divergence. The number of clusters per plant, number of pods per cluster, and number of pods per plant showed a significantly positive association with grain yield. G × E analysis contributed to the detection of high-yielding blackgram mutants, and mutagenesis caused by gamma rays helped discover functional mutants such as G13, G7, and G34 that are promising for environments $E_1$, $E_2$, and $E_3$, respectively. High-yielding mutants for specific environments should be further evaluated in larger plots, as they could be used for recombination breeding to obtain desirable segregants for sustainable blackgram production.

**Supplementary Materials:** The following are available online at https://www.mdpi.com/article/10.3390/agronomy11071287/s1. Table S1. Weather parameters during crop growth stages; Table S2. Estimates of mean values, range, broad sense heritability (h2b %) for eight yield-related traits in the M2 generation of blackgram cultivars viz., ADT3, CO6 and TU 17-9 (continued); Table S3. Estimates of mean values, range, broad sense heritability (h2b %) for eight yield-related traits in the M3 generation of blackgram cultivars viz., ADT3, CO6 and TU 17-9 (continued); Table S4. Contribution of varying gamma doses/irradiation causing distinct morphological mutations in M2 generation; Table S5. Contribution of varying gamma doses/irradiation causing different morphological mutations in M3 and M4 generations.

**Author Contributions:** Conceptualization: S.G., N.M., and D.S.; methodology: M.D., S.G., N.M., and D.S.; validation: M.D.; formal analysis: M.D. and A.K.; investigation: M.D.; resources: S.G., D.S., and N.M.; data curation: N.M. and S.G.; writing—original draft preparation: M.D. and A.K.; writing—review and editing: A.K., M.D., S.G., N.M., and D.S.; supervision: N.M. and S.G.; project administration: S.G.; funding acquisition: S.G. All authors have read and agreed to the published version of the manuscript.

**Funding:** This research received funding from the Board of Research in Nuclear Sciences (BRNS), Government of India (GoI).

**Institutional Review Board Statement:** Not applicable.

**Informed Consent Statement:** Not applicable.

**Data Availability Statement:** Data are available from the authors upon request.

**Acknowledgments:** The authors wish to acknowledge the Board of Research in Nuclear Sciences (BRNS), Government of India, for financial support rendered and the Tamil Nadu Agricultural University for provided the experimental fields for executing research trials. The data presented are part of the Ph.D. thesis of M.D. supervised jointly by N.M. and S.G.

**Conflicts of Interest:** The authors declare no conflict of interest.

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
