# Peer review of "Development of Novel Blackgram (Vigna mungo (L.) Hepper) Mutants and Deciphering Genotype × Environment Interaction for Yield-Related Traits of Mutants"

_agronomy, doi:10.3390/agronomy11071287_

Round 1
Reviewer 1 Report
Were traits all measured on an individual plant basis? This is open to criticism that plant to plant competition influences phenotype both from differences in plant at spacing (Both within and between rows), and genetic differences between plants in growth characteristics. Plot means over at least 10 plants per plot would be better. 25-30 plant per plot desirable, and data aught to be based on plot means with 3-4 replication per site.
By M5 the level of plant heterogeniety within lines would be low, and characterisation would be more reliable to describe genetic differences. Thus GXE analyses across environments at this level would be suitable.
It is not clear whether all levels of radiation (doses) contributed to the selection of mutants, nor whether the doses were compared for selection outcomes. This analysis should have been reported in this paper.
No report was given to show comparison of mutants versus the original parents, and hence the gains in selection.
It is not clear how the selection of mutants was conducted. Was the selection index uniform across the varietal mutant populations? were yield per plant and each component trait, and earliness, equally weighted? (Yield traits would be interrelated but subject to different levels of heritability) or were outstanding lines for any one trait selected as well?
There was indeed differences in GXE. Such differences may be expected too across the target environments on-farm. This study can serve as a preliminary to future state-wide and national trials. Adaptation of individual mutants to specific test environments (3 locations) is not relevant to the bigger picture state-wide, especially as results are confined to 1 year of a GXE trial. But measurement of stability does provides data to justify further testing. There are limited inferences that can be drawn from this trial, as just the first step towards much wider evaluation.
The variation form the selected mutants should be compared against the variation form conventional breeding, perhaps combined for comparison in state wide trials.
It is hard to know whether radiation generated more diversity than would be obtained in conventional breeding. At least a comparison with existing varieties would be interesting. radiation is one breeding method, but how does it compare with other breeding methods?
Reviewer 2 Report
This study of blackgram in India involves 36 mutants and 3 environments, but unfortunately only a single year so repeatability across years cannot be assessed. Some agronomy journals require that field studies be repeated over 2 or 3 years, but I do not know the requirements of this particular journal. The statistical analysis is badly flawed, but performing a much better statistical analysis would be a small effort compared to the effort involved in collecting the data.
Line 40 mentions two market classes: large black seeds and early maturity, or small greenish seeds and late maturity. For comparison, studies of dry beans ordinarily segregate yield trials into market classes so that genotypes compete only within their own market class. The authors should explain more carefully how market classes have, or have not, been incorporated their approach to statistical analysis.
Line 177 mentions that “most of the stable genotypes might not always have good yield efficiency,” and indeed a negative correlation between stability and yield is the rule. But the direct implication of this fact does not seem to have been understood here, namely that it makes no sense to “integrate yield and stability in a single index.” It would be better to delete this GSI index, and focus instead on which genotypes win where.
Table 1 includes skew and kurtosis, which I have never before seen reported for a yield trial. The authors might provide the readers with an explanation of why these was included and how to use or interpret this information.
Table 4 provides the F-value, but this value should be followed with a probability.
Line 355 discusses “ideal” genotypes (and again line 449). But “ideal” is defined as high stability and high yield, whereas this manuscript has already emphasized that one cannot get both. So wanting an “ideal” genotype is like wanting a unicorn instead of a horse – it just does not exist. This discussion of “ideal” genotypes seems fundamentally misguided and unhelpful.
Figure 4a and 4B identify the graphs as AMMI graphs, but they look like GGE graphs. Which are they? Also, these graphs are illegible; they need a larger font to identify the genotypes and environments.
Line 438 discusses the which-won-where pattern in a graph. But which-won-where from a graph should also be compared with the actual data in Table 5. Note that G31 is the winner in E1, G7 in E2, and G34 in E3 according to the actual data.
Round 2
Reviewer 1 Report
I am satisfied that the authors have have adequately addressed reviewer comments.
Reviewer 2 Report
This is my second review of this manuscript. It has been improved considerably. I have just three small comment.
My first review ended with advice to compare winners from graphs with the actual data in Table 5, and I continue to suggest that. Note that G13 is the winner in E1 (not G31 which was a typo in my first review), G7 in E2, and G34 in E3. For comparison, the final conclusion focuses on G18 and G29, which never win in the actual data. Why not give more emphasis on the actual winners?
Figure 4a clearly shows a contrast in interaction patterns between E3 at the top and E1 at the bottom, with E2 closer to the middle. Can you give the contrast between E3 and E1 a plausible environmental interpretation – wet versus dry, hot versus cold, different soils, or whatever?
In order to increase the interest for readers located outside India, can you discuss the extent to which your results are likely to be useful for breeders in other countries? Are you willing to share germplasm with other researchers?
